# Chemical Composition of Earthworm (*Dendrobaena veneta* Rosa) Biomass Is Suitable as an Alternative Protein Source

**DOI:** 10.3390/ijerph20043108

**Published:** 2023-02-10

**Authors:** Mariola Garczyńska, Joanna Kostecka, Grzegorz Pączka, Anna Mazur-Pączka, Tomasz Cebulak, Kevin R. Butt

**Affiliations:** 1Department of the Basis of Agriculture and Waste Management, Institute of Agricultural Sciences, Land Management and Environmental Protection, College of Natural Sciences, University of Rzeszow, ul. Cwiklinskiej 2, 35-601 Rzeszow, Poland; 2Department of Food Technology and Human Nutrition, Institute of Food Technology and Nutrition, College of Natural Sciences, University of Rzeszow, 4 Zelwerowicza Str., 35-601 Rzeszow, Poland; 3School of Natural Sciences, University of Central Lancashire, Preston PR1 2HE, UK

**Keywords:** food challenges, earthworm, *Dendrobaena veneta*, chemical composition, amino acids, fatty acid profile, organic wastes, vermiculture

## Abstract

The selected chemical composition (dry matter, ash, total protein, and crude fat) of the integumentary muscles of *Dendrobaena veneta* were determined, plus the dry matter (DM) percentage content of 17 amino acids and the profile (%) of fatty acids. Results were compared with a more fully studied earthworm, *Eisenia fetida*. In addition, the composition of exogenous amino acids was compared to the WHO standard for pork, beef, and chicken eggs. Both earthworm species were grown on the same kitchen waste, and protein composition was analyzed using the same methods. Studies indicated that the muscle of *D. veneta* was characterized by a high level of protein (76.82% DM). A similar content of exogenous amino acids was observed in the protein of both earthworms, but for phenylalanine and isoleucine, slightly higher levels were recorded for *E. fetida*. More histidine, lysine, threonine, isoleucine, and arginine were found in earthworms compared with chicken egg white. Fatty acids play an essential role in balancing human or animal feed and their content determines the dietary and nutritional value of the food. Both earthworm species contained the appropriate content of saturated and unsaturated acids. In *D. veneta*, a higher content of arachidonic acid was found, and in *E. fetida*, lauric, tridecanoic, and palmitic acids were present. Future issues of food security may force us to seriously consider earthworm protein for indirect or even direct human consumption.

## 1. Introduction

Sustainable food production is a priority in agriculture, along with the protection of agroecosystems, and the awareness of public health. Food systems require the development of sustainable agricultural practices, balanced distribution systems, the creation of a sustainable diet, and a reduction in food waste. It is estimated that by 2025, the world’s population will reach over 8 billion, which is likely to be associated with food insecurity [1,2].

According to the McGuire [3], food production may double by 2030; however, factors limiting this include a constant decrease in area of agricultural land and a reduction in global freshwater resources, which makes animal breeding more difficult due to higher maintenance costs [2]. To satisfy the nutritional needs of the human population, a search for less conventional sources of nutrients is required, as in 2019, protein of animal origin provided only 55% of the total human needs in Europe [4,5,6,7,8,9]. By contrast, the intensification of animal production has led to the search for a less expensive feed of the highest quality. A promising alternative to vertebrate protein is that obtained from invertebrates (e.g., from the integumentary muscles of earthworms or from some insect species) [10,11]. These alternative protein sources have great potential as dietary components due to their high nutritional value, primarily the complete protein and fat content. The major factor limiting this source as food for humans is the lack of acceptance in many European countries and its limited availability. Current and future trends are moving towards the search for “superfoods”, and earthworm biomass is nutritionally attractive. Recipes for dishes with the inclusion of earthworms were already known in the 1970s [12]. The problem is with non-acceptance of this alternative food. Conti et al. [13] and Verneau et al. [14] indicate that the best solution to this problem may be the invisible inclusion of invertebrates (earthworms and insects) in food products.

Earthworm biomass has interesting nutritional properties, but in order to be commercialized as a product for human consumption and/or as animal feed, it must be safe for the final consumer. To ensure appropriate safety standards, the finished alternative food product should undergo a hygienic and sanitary assessment and it is necessary to assess the microbiological profile of the food, process hygiene indicators and additionally examine the chemical profile for possible presence of pesticide residues and toxic elements [15].

Conti et al. [16], analyzed the microbiological composition of fresh earthworms *Eisenia fetida* and earthworm meal (freeze-dried and dried, produced from fruit and vegetable waste). All samples showed the absence of *Salmonella* spp. and *Listeria monocytogenes*, in accordance with Regulation (EU) 2073/2005 [17]. Studies have shown that both fresh earthworms and earthworm meal can be a safe food in terms of microbiological criteria. As in the case of the production of other edible terrestrial invertebrates, appropriate technological treatment, packaging, and storage conditions should be ensured to prevent microbiological contamination. Therefore, an important step to provide consumers with safer products should be the establishment of specific guidelines for the production and commercialization of earthworms if they are to be bred for human consumption.

Considering the continuous growth of the global human population, the demand for nutrients, including complete protein, will also increase [6,9].

There are uncertainties in the world related to the number of known species of earthworm. Sims and Gerard [18] recognized 3000 species, Sinha et al. [19] reported 4400, while Grdiša et al. [20] reported that 7000 species had been recorded. Approximately 40 of these species have been described in Poland, of which some, e.g., *D. veneta* or *E. fetida*, can be easily bred as they are used in the vermicomposting process of selected organic waste [21,22,23]. This process allows the transformation of segregated organic waste into biofertilizer-vermicompost, which can form a part of the circular economy [24]. However, the earthworm biomass produced can also be used as an alternative food source for animal and even human nutrition [7]. In studies evaluating the acceptance of rice wafers, either pure or enriched with earthworm protein, it transpired that social acceptance was mainly based on the way the information was presented on the label. Distinguishing high health and nutritional values was a factor that encouraged consumers to buy this product, which was particularly visible among men, while women behaved more conservatively. We are at the beginning of the journey to introduce new foods, and it seems that there is a certain consumer acceptance of such products. More time and appropriate actions are needed to further convince consumers of the naturalness of products enriched with invertebrate proteins [25].

Numerous studies indicate the possibility of using the biomass of selected species of earthworms obtained from vermiculture (*E. fetida*, *Eisenia andrei*, *Peronyx excavatus*, *Eudrilus eugenigae* or *D. veneta*) for fish nutrition [26,27,28], broilers [29,30] or pigs [31,32]. It should be noted that earthworm biomass forms a natural, valuable dietary component for numerous wild mammals, including red fox (*Vulpes vulpes*), raccoon (*Procyon lotor*), European badger (*Meles meles*) and brown bear (*Ursus arctos*) [33,34].

The current study aimed to determine selected nutritional indicators of *D. veneta* (amino acid and fatty acid profile), grown on kitchen waste, compared with those of *E. fetida*, in addition to the composition of their essential amino acids in relation to the WHO standard [1] for pork meat, beef and chicken eggs, and thus to assess the suitability of earthworm protein as a potential option for animal and/or human consumption.

## 2. Materials and Methods

### 2.1. Material

The material used in the study consisted of *Dendrobaena veneta* (Rosa 1886) obtained from a long-term breeding program conducted at the Department of the Basis of Agriculture and Waste Management at the University of Rzeszów. This species is of Holarctic origin and originally inhabited Asia Minor and the Caucasus, from where it was brought to an artificially created environment rich in organic matter [35]. It is characterized by a short life cycle, where a complete cycle takes approximately 100–150 days, with sexual maturity reached after approximately 65 days [36] (Figure 1).

### 2.2. Laboratory-Scale D. veneta Biomass Production

Earthworm biomass production was carried out in vermireactors (n = 5) constructed of plastic containers with dimensions of 200 × 150 × 150 mm (length × width × height) (Figure 2) [37]. The base of each was perforated with small holes to drain the excess liquid. Each vermireactor was placed in a larger box so that their bases were not in contact (30 mm distance between them) to store excess leachate. Kitchen waste including boiled pasta (150 mL), bread (150 mL), potato (150 mL), and apple waste (150 mL) (600 mL in total) mixed with 300 mL of cellulose were placed in each vermireactor, and its remaining capacity was filled with a substrate in the form of a horticultural soil universal substrate for all ornamental plants, Floro-hum: pH 5.5–6.5, composed of highmoor peat, lowmoor peat, perlite, sand, microelements, and mineral fertilizer NPK [38]. Fifty individual, sexually mature earthworms were introduced into each vermireactor. Cultures were secured with a nylon mesh, which prevented earthworm escape, covered with cardboard to prevent the substrate from drying out and kept in a climatic chamber at 20 ± 0.5 °C [39].

To maintain appropriate humidity within the vermireactors, they were wetted every 10 days with the same amount of tap water (pH—7.5, conductivity—486 μS cm^−1^, nitrates (V)—8.5 mg·dm^−3^, nitrates (III)—0.01 mg·dm^−3^, Mg—14.2 mg·dm^−3^, water hardness—231 CaCO_3_ mg·dm^−3^). Analytical results were presented as the means and SD of five replicates.

### 2.3. Determination of Earthworm Chemical Composition

After a period of eight days, adult earthworms were selected from the vermireactors and examined for their chemical composition. After the contents were removed from their digestive tract, the bodies were assayed with the use of standard methods of fodder analysis given by AOAC 2003 [40] to identify the following parameters:–water content (according to PN-ISO1442, which involves drying the sample in an Ecocell laboratory dryer from BMT at a temperature of 103 ± 2 °C to obtain dry matter);–total ash (according to a method complying with PN-ISO936, which involves drying the analyzed sample, to be subsequently incinerated in a Snol muffle furnace at a temperature of (550 ± 25) °C, and after cooling down the mass of the residue is determined;–total nitrogen (with the Kjeldahl method in compliance with PN-75 A-04018, with conversion to protein);–fat (with the Soxhlet method in a Kjeltec 2200 apparatus manufactured by Boss. Before this, the samples were subjected to hydrolysis with hydrochloric acid);–contents of amino acids by Zeng et al. [41] (by hydrolyzing the sample with 6M HCl for 24 h at a temperature of 110 °C and rinsing with 0.1 molar solution of HCl and distilled water; the hydrolysate was then evaporated, and the residue was dissolved in a buffer of pH 2.2; the contents of amino acids were determined with the use of AAA-400 amino acid analyzer, which performs an assay based on liquid chromatography—following separation in the column, amino acids react with ninhydrin. The sulfur-containing amino acids were subjected to oxidizing hydrolysis with formic acid and hydrogen peroxide and then examined with AAA-400);-profile of fatty acids by Zhang et al. [42] (the samples were prepared following the Folch method (extraction with chloroform-methanol (2:1) mixture, methylation BF_3_/methanol). The profile was examined with a Varian 3400CX gas chromatograph, equipped with a flame ionization detector (FID), with the use of a CP-WAX column (length 50 m, diameter 0.53 mm); conditions of chromatograph operation: carrier gas—argon, the temperature of the dispenser −200 °C, temperature of the detector −240 °C, temperature of the column −60–220 °C). The analytical results of the fatty acid profiles were used to calculate the sum of saturated and unsaturated acids in fat. The content of amino acids and the profile of fatty acids in the fat of earthworms were determined in the material previously subjected to the freeze-drying process. Due to a lack of specific information on the chemical composition of *D. veneta*, research was undertaken to determine the content of protein and amino acids and the profile of fatty acids in the integumentary-muscular sacs as a possible raw material for use in product compositions for human and livestock nutrition. Thereafter, the composition of selected indicators of nutritional quality of integumentary-muscle sacs of *D. veneta* were compared with *E. fetida*, an earthworm for which much more information is available.

## 3. Results

### Chemical Composition of the D. veneta Earthworm

Selected chemical components (dry mass, raw ash, total protein, and raw fat) of the body of worms cultivated on kitchen waste are shown in Table 1. Earthworms are characterized by high water content in the fresh mass. Therefore, the possibility of commercial use of integumentary-muscle sacs may depend on using them in a dried form, from which a concentrated source of nutrients can be obtained. Dry matter of *D. veneta* was determined to contain 5.17% ash, 76.82% protein, and 12.21% fat. The total protein content in the dried integumentary-muscle sacs of earthworms reached a level close to that of commercial protein concentrates.

The factors examined also included 17 endogenous and essential amino acids (Table 2) and the profile of fatty acids (Table 3). In the body of *D. veneta*, a beneficial presence of amino acids was found, which in the dried raw material corresponded to the nutritional value of a hen’s egg, as evidenced by a CPSP (hydrolyzed protein) index of 1.1.

Exogenous amino acid content diversity applies to all protein sources, including the protein derived from *D. veneta*. In the tested material, higher contents of exogenous amino acids were found in hen’s egg white in the case of phenylalanine, histidine, lysine, threonine, isoleucine, and arginine. In the case of protein derived from *D. veneta*, compared to the protein from *E. fetida*, we can only relate a higher content of the amino acids phenylalanine and isoleucine.

The dietary value and nutritional value in balancing food for humans or animal feed is also determined by the appropriate content of fatty acids (Table 3). In *D. veneta*, a higher content of arachidonic acid was found, and in *E. fetida*, the following acids were found: lauric, tridecanoic, and palmitic acids. In the case of monounsaturated fatty acids, two showed differentiations in both species—C_16:1_ palmitoleic acid (>*E. fetida*) and C_22:1_ euric acid (>*D. veneta*). The percentage of total polyunsaturated fatty acids in both species was similar, except for docosapentaenoic and clupanodonic acids in the bodies of *D. veneta*.

## 4. Discussion

Apart from vermicompost, earthworm biomass is also produced during vermicomposting and is attractive as it has a high nutritional value due to its protein content. As earthworms contain a low percentage of dry matter, they are classified as succulent fodder, with test results indicating that the dry matter content depended on the species (*D. veneta* (fed with kitchen residues—12.8% DM), *E. fetida* (fed with kitchen residues—17.8% DM; fed with cattle manure—18.2% DM)) [43,45]. For nutritional reasons, the amount of protein is essential, and exogenous amino acids mainly determine its value. Due to their content, earthworms are classified as high-protein feeds comparable with fish, meat, and bone meal [7,45,46].

Results indicated that high nutritional values characterize the protein of earthworms; the essential amino acid index CSPS% in *D. veneta* was 1.1 and so slightly exceeded the CSPS value for chicken egg white. However, it was 0.2 lower than muscle protein from *E. fetida*. In the study of Kostecka et al. [43], *E. fetida* and *D. veneta*, were fed kitchen leftovers and were characterized by a high content of essential amino acids. In the present comparison, it has been shown that the content of some exogenous amino acids differed within these two species. For example, in *D. veneta*, the content of phenylalanine and isoleucine were increased by 0.6 and 0.1% DM, respectively. Many authors have already drawn attention to the high content of exogenous amino acids, such as lysine, methionine, cystine, tryptophan, and threonine, in earthworm muscle [31,43,44,47].

According to Antonova et al. [7], the most crucial animal feed component is lysine, a combination of the amino acids methionine-cysteine and phenylalanine-tyrosine. No differences were found between *D. veneta* and *E. fetida* in lysine content, and content of the listed essential amino acids in *D. veneta* was high. Therefore, the dried mass of earthworm integumentary muscle sacs can be used as an animal feed ingredient. However, their use as a human food ingredient may raise some concerns due to the low nutritional acceptance of this type of product, especially in the culinary culture of European countries and North America. Nevertheless, for ethical and health reasons, the current trends point towards food with high health quality. Available research shows that one way to introduce such products for food purposes is to introduce them invisibly as one of the ingredients in a food product [13,14]. This type of solution is supported by Tedesco et al. [15], who showed that the meal produced from earthworm integumentary sacs is safe in microbiological and chemical terms (lack of pesticides and heavy metals), as a result of the use of kitchen waste, i.e., the raw material for earthworm culture has already passed the quality requirements for safe food.

Our studies also determined the fatty acid profile of *D. veneta*, as these perform many functions in the body, and a deficiency can cause many disorders. Numerous publications emphasize their great importance in human and animal nutrition, especially unsaturated acids [48,49,50]. The results obtained in this study indicated that the unsaturated acids in the fat of earthworms account for 54.30%, and the saturated acids account for 33.40%. The highest amount of unsaturated acids was oleic acid (30.70%), and the largest share of saturated acids was stearic acid (6.7%). Earthworm tissues contain predominantly long-chain fatty acids, which nonruminating mammals cannot synthesize, so earthworms can be an excellent dietary supplement. Determination of the fatty acid profile and the ratio of unsaturated to saturated acids is essential for the construction of a balanced diet for livestock [45].

Saturated acids perform a reserve and energy function, while unsaturated acids participate in metabolic processes, forming prostaglandins and prostacyclins. In the body of the tested *D. veneta* earthworms, the beneficial presence of many valuable saturated acids was found, e.g., palmitic acid and eicosanoic acid (arachidic acid); monounsaturated acids, e.g., oleic acid; and polyunsaturated acids, e.g., linoleic and eicosadienoic acids (Table 4). In both earthworm species, significant polyunsaturated fatty acids were also found—linoleic (C_18:2_) and linolenic (18:3) acids, which have a significant impact on human and animal health, and are also precursors in the synthesis of arachidic acid [50,51,52].

Using earthworm biomass obtained from vermiculture for animal nutrition is an opportunity to reduce the costs of their production thanks to vermicomposting technology. This may work in organic farms, where the excess biomass of *E. fetida* and *D. veneta* earthworms can be used to feed poultry, pigs, or fish. At the same time, periodic monitoring of the chemical composition of earthworms is needed because this may show variability depending on the food consumed or other abiotic factors. Major factors determining the use of earthworm biomass are the production volume and costs [43,45].

As far as human nutrition is concerned, attempts have also been made in this regard, but this applies mainly to Asian and South American countries [53,54,55]. In some countries, earthworms supplement the human diet [55]. For example, locally in China, people have been using earthworms as food for several centuries, and in Taiwan, Henan, and Guangdong provinces, unique dishes based on earthworms are still prepared. Meals prepared from earthworms can often have a better protein content and amino acid composition than that in fish or soybean meals [56].

Earthworms from the family Glossoscolecidae are, among others, an essential component of the diet of the Ye’Kuana Amerindian tribe, as they contain 70% protein [54,57]. It is confirmed by the average consumption of *Andiorrhinus kuru* and *Andiorrhinus moto* in Venezuela, which amounts to 1.7–2 kg per person per year. In the set of fatty acids of these earthworms, the beneficial arachidonic acid has a considerable proportion [54].

The factor limiting human consumption of food based on earthworms is the lack of acceptance in many European countries, which is why it is difficult to set rules for preparing and balancing a diet that includes these invertebrates. However, it is difficult to underestimate this possibility in times of food crisis and when seeking alternative food. After all, invertebrates may be an essential factor in solving the problem of global hunger and malnutrition [58].

## 5. Conclusions

The results indicated that the body of *D. veneta* is rich in nutrients, including available protein. Nevertheless, a critical factor for establishing a new trend in human and livestock nutrition is an appropriate content of exogenous amino acids. In the tested earthworms, a favorable content of these was found, which in the dried raw material corresponded to the nutritional value of a chicken egg, as evidenced by a CPSP index of 1.1. The second factor that could affect the proper dietary balance is fatty acids content, and the appropriate level of saturated, monounsaturated, and polyunsaturated fatty acids characterizes the tested earthworms.

Earthworm biomass fed on segregated kitchen waste could be used in human nutrition and as animal feed on an organic farm because, in times of food crisis, it is important to search for alternative sources of good quality food. The goal of producing commercial preparations based fully or partially on earthworms may soon be crucial and justified. Therefore, shaping a change in eating habits among Europeans and implementing the described animal nutrition preparations on an industrial scale may be a significant challenge in the coming years. Nevertheless, earthworm protein could certainly prove to be a real alternative source for animal feed and, possibly, for human consumption.

## Figures and Tables

**Figure 1 ijerph-20-03108-f001:**
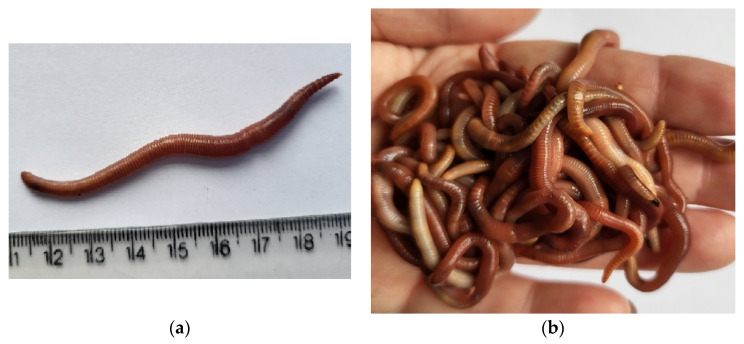
(**a**) A single adult *Dendrobaena veneta*; and (**b**) as group taken from a bed for processing organic material.

**Figure 2 ijerph-20-03108-f002:**
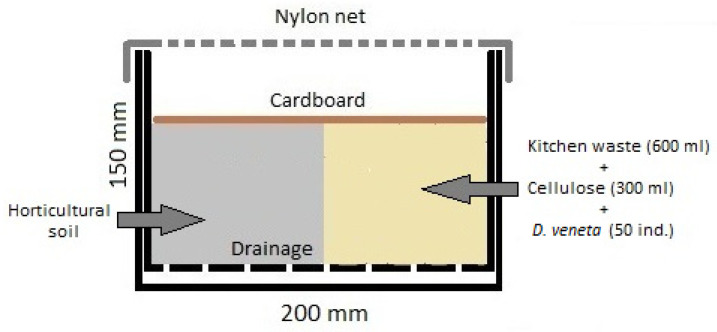
Schematic of a vermireactor, modified from Pączka et al. [37].

**Table 1 ijerph-20-03108-t001:** Chemical composition (%) of *D. veneta* cultivated in kitchen waste.

Traits	*D. veneta* Content
	in Dry Matter
dry mass	100.0 ± 0.02
crude ash	5.17 ± 0.01
total protein	76.82 ± 0.38
crude fat	12.21 ± 0.57

**Table 2 ijerph-20-03108-t002:** Content of amino acids (% DM) in the biomass of *D. veneta* cultivated on kitchen waste.

Amino Acids	*D. veneta*	*E*. *fetida* *	Beef **	Pork **	Pattern Protein Egg **
	Endogenous amino acids				
Aspartic acid (Asp)	8.22 ± 0.52	5.75 ± 2.03			
Serine (Ser)	3.83 ± 0.39	2.89 ± 0.85			
Glutamic acid (Glu)	8.94 ± 1.13	7.12 ± 1.72			
Proline (Pro)	2.55 ± 0.21	1.94 ± 0.15			
Glycine (Gly)	3.50 ± 0.69	2.59 ± 0.69			
Alanine (Ala)	3.48 ± 0.69	2.62 ± 0.69			
Cysteine (Cys)	0.84 ± 0.10	0.64 ± 0.18			
Tyrosine (Tyr)	2.16 ± 0.38	1.43 ± 0.39			
	Essential and semi-essential amino acids (EAA)				
Phenylalanine (Phe)	2.91 ± 0.40	2.3 ± 0.25	1.8	1.7	2.6
Histidine (His)	1.62 ± 0.29	2.1 ± 0.28	1.5	1.7	1.2
Lysine (Lys)	4.74 ± 0.51	4.9 ± 0.38	3.7	3.9	3.4
Methionine (Met)	0.80 ± 0.08	1.1 ± 0.20	1.2	1.1	1.5
Threonine (Thr)	2.34 ± 0.20	2.8 ± 0.43	1.7	2.0	2.2
Valine (Val)	3.68 ± 0.44	3.8 ± 0.53	2.2	2.4	3.1
Tryptophan (Trp)	—	—	0.2	0.5	0.8
Isoleucine (Ile)	3.05 ± 0.52	2.9 ± 0.41	2.0	2.0	2.5
Leucine (Leu)	3.66 ± 0.45	6.2 ± 0.39	3.5	3.5	4.3
Arginine (Arg)	5.16 ± 0.45	6.0 ± 0.33	2.9	2.7	3.2
Total EAA	27.96	32.1	20.7	21.5	24.8
CPSP (%)	1.1	1.3	0.8	0.9	1

* Kostecka et al. [43] ** Ding et al. [44].

**Table 3 ijerph-20-03108-t003:** Content of fatty acids (%) in *D. veneta* cultivated in kitchen waste.

Fatty Acids Profile (% in Total Acids)		
Fatty Acids	*D. veneta*	*E. fetida* *
saturated fatty acids		
Lauric acid C_12_	2.67 ± 0.16	3.60 ± 0.18
Tridecanic acid C_13_	0.16 ± 0.02	0.27 ± 0.04
Myristic acid C_14_	5.41 ± 0.15	5.64 ± 0.13
Pentadecanoic acid C_15_	0.29 ± 0.03	0.25 ± 0.03
Palmitic acid C_16_	16.19 ± 0.2	17.06 ± 0.12
Heptadecanoic acid _C17_	0.51 ± 0.04	0.50 ± 0.02
Stearic acid C_18_	6.74 ± 0.22	6.45 ± 0.61
Kwas arachidowy C_20_	0.16 ± 0.01	0.12 ± 0.01
Behenic acid C_22_	1.09 ± 0.04	0.96 ± 0.06
Lignoceric acid C_24_	0.13 ± 0.01	0.12 ± 0.01
Total saturated acids	33.38	35.31 ± 0.64
monounsaturated fatty acids		
Myristoleic acid C_14:1_	0.26 ± 0.02	0.33 ± 0.05
C_15:1_	0.15 ± 0.01	0.13 ± 0.01
Palmitoleic acid C_16:1_	4.77 ± 0.87	5.60 ± 0.76
C_17:1_	0.68 ± 0.12	0.79 ± 0.05
Oleic acid C_18:1_	30.68 ± 1.14	31.05 ± 0.48
Eicosenoic acid C_20:1_	1.97 ± 0.28	1.88 ± 0.32
Euric acid C_22:1_	1.15 ± 0.16	0.56 ± 0.33
Total monounsaturated acids	39.67	40.860 ± 0.678
polyunsaturated fatty acids		
C_14:2_	1.03 ± 0.07	1.24 ± 0.13
Linoleic acid C_18:2_	9.51 ± 0.17	9.77 ± 0. 42
Linolenic acid C_18:3_	1.21 ± 0.16	1.12 ± 0.17
Eicosadienic acid C_20:2_	0.70 ± 0.04	0.77 ± 0.08
Arachidonic acid C_20:4_	1.17 ± 0.7	1.24 ± 0.09
Docosapentaenoic acid C_22:5_	0.246 ± 0.03	0.13 ± 0.01
Clupandonic acid C_22:6_	0.730 ± 0.0	0.20 ± 0.04
Total polyunsaturated acids	14.62	14.487 ± 0.834
Unmarked	10.02	9.645 ± 1.025

* Kostecka et al. [43].

**Table 4 ijerph-20-03108-t004:** Function and amount of selected fatty acids in the fat of composting earthworms (%).

Fatty Acid	Function	*D. veneta*	*E. fetida*
palmitic acid	activator of many enzymes	16.19 ± 0.2	17.06 ± 0.12
oleic acid	increases the immunity of animals and prevents cardiovascular diseases	30.68 ± 1.14	31.05 ± 0.48
linoleic acid	participates in the formation of prostaglandin PGE_1_ and prostacyclin, which regulate the tone of the walls of blood vessels	9.51 ± 0.17	9.77 ± 0.42
eicosadienoic acid_:_	is part of omega-3 fatty acids, contributing to the increase in immunity	0.70 ± 0.04	0.77 ± 0.08

## Data Availability

The data presented in this study are available on request from the corresponding author.

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
