# Peer review of "Chemical Composition of Earthworm (Dendrobaena veneta Rosa) Biomass Is Suitable as an Alternative Protein Source"

_ijerph, 2023, doi:10.3390/ijerph20043108_

Round 1

Reviewer 1 Report

This is an excellent manuscript. It is very well-written and contains numerous well-cited references. The researchers conducted a thorough analysis of the different components of the two species of earthworms, including protein content, fatty acids, and amino acid composition. The authors provided useful recommendations for using these earthworm species for animal feed and human nutrition. Two minor comments: 
1. "Sustainable" was used three times in the second sentence. Perhaps other words could be substituted?
2. Line 94: perlite not pearlite

Author Response

Response to Reviewer 1  Comments on Ref.: Ms. No.– IJERPH 2129789

Earthworm (Dendrobaena veneta Rosa) biomass as an alternative protein source

Please see our responses to the points raised by the reviewer below.

Reviewer 1

This is an excellent manuscript. It is very well-written and contains numerous well-cited references. The researchers conducted a thorough analysis of the different components of the two species of earthworms, including protein content, fatty acids, and amino acid composition. The authors provided useful recommendations for using these earthworm species for animal feed and human nutrition.

Thank you for these comments

Two minor comments: 

  1. "Sustainable" was used three times in the second sentence. Perhaps other words could be substituted?

Thank you. Changed

  1. Linia 94: perlit, no pearlit

Thank you. Changed

Reviewer 2 Report

This manuscript presents data on earthworm biomass as an alternative protein source in times of food crisis. As I am not a subject matter expert on earthworm biomass, there were a few items, that if clarified, would help my understanding and perhaps help other readers as well.

* It was indicated that the worms were sexually mature before introducing the worms to the vermireactor and kitchen waste. Can the authors clarify if there is any significance to using only sexually mature worms?  Why not use worms earlier in the life cycle?  What was the diet content prior to sexual maturity?

* How many days after introducing the worms to the vermireactor were the worm contents analyzed.  Is there any significance to this number?  Could the contents have been analyzed immediately upon receipt from the Department of the Basis of Agriculture and Waste Management?

*If the premise for the study is to identify a protein source in times of food crisis, then what's the significance of choosing kitchen waste for earthworm biomass production verses manure? (Is kitchen waste expected to be scarce in times of food crisis)?

*Of the thousands of different species of earthworms, what's the significance of comparing D.veneta to E.fetida? 

*How sensitive is the variability of chemical composition from the earthworm biomass?  Which specific factors (other than, of course, the type of feed) might alter the biomass content?

*The authors suggest that "the factor limiting humans' consumption of food -based earthworms is lack of acceptance." To support this statement, it might be helpful if the authors could describe (in the background or introduction) outcomes of an animal study where earthworms replaced or substantially substituted for the normal protein source.

Author Response

Response to Reviewer 2  Comments on Ref. :Ms. No.– IJERPH 2129789

Earthworm (Dendrobaena veneta Rosa) biomass as an alternative protein source

Please see our responses to the points raised by the reviewer below.

Reviewer 2

* It was indicated that the worms were sexually mature before introducing the worms to the vermireactor and kitchen waste. Can the authors clarify if there is any significance to using only sexually mature worms?  Why not use worms earlier in the life cycle?  What was the diet content prior to sexual maturity?

The age structure of earthworms includes mature individuals with a developed saddle (clitellum), immature individuals and cocoons. Mature specimens were added to the vermireactors, because at this stage of development earthworms are capable of reproduction, which means that the population of these invertebrates can be expected to develop at a faster pace. Using immature specimens, the Lumbricida biomass production process is significantly extended, and besides, immature D. veneta is more sensitive to numerous habitat changes. In similar studies on the biochemical analysis of earthworms, mature specimens are added. An interesting research aspect in the future will be the comparison of the chemical composition of biomass in mature and immature individuals.

 Before puberty (from cocoon to puberty) earthworms were fed exclusively on kitchen waste.

* How many days after introducing the worms to the vermireactor were the worm contents analyzed.  Is there any significance to this number?  Could the contents have been analyzed immediately upon receipt from the Department of the Basis of Agriculture and Waste Management?

Thank you. Added

After removing the earthworms from the vermireactors, their integumentary and muscle sacs were cleaned on tissue paper, on average 8 days. Immediately after their purification, the D.veneta biomass was subjected to chemical analysis.

*If the premise for the study is to identify a protein source in times of food crisis, then what's the significance of choosing kitchen waste for earthworm biomass production verses manure? (Is kitchen waste expected to be scarce in times of food crisis)?

In times of food crisis, various organic wastes must be taken into account. D.veneta earthworms can transform a lot of waste, they are also used to transform manure, everything depends on the possibility of obtaining the initial biomass. Currently, an important problem is the waste of food on a fairly large scale, therefore a lot of kitchen waste is generated, which can be environmentally friendly managed through vermicomposting, and the obtained earthworm biomass can be used as food for animals or it is also possible to use it in human nutrition. In addition, it can be assumed that properly selected kitchen waste will not be toxic to Lumbricidae, will not contain excessive content of various stressors, e.g. pesticides or heavy metals. In the era of the circular economy, we should stop throwing away food leftovers, which are a valuable organic resource, so the solution to this problem is greater environmental awareness of consumers.

*Of the thousands of different species of earthworms, what's the significance of comparing D.veneta to E.fetida? 

Earthworms live in different ecomorphological layers, which depends on their ecology. Most of the species of earthworms cannot occur in a large accumulation of organic waste, they occur in the mineral layer of the soil, therefore obtaining biomass on a different scale would be difficult. The most common earthworm species found in natural habitats is Lumbricus terrestris. This species is distinguished by a relatively slow reproduction process, it lives mainly underground and usually appears on its surface when there is high humidity. On the other hand, both epigeic species Eisenia fetida and Dendrobaena veneta are commonly used for vermicomposting of organic waste in temperate climate conditions. Representatives of these species are characterized by a fast rate of reproduction and high resistance to changing environmental conditions (Dominguez et al. 2011), which results in a fairly high possibility of obtaining their biomass at a rapid pace.

*How sensitive is the variability of chemical composition from the earthworm biomass?  Which specific factors (other than, of course, the type of feed) might alter the biomass content?

The experiment was carried out under controlled conditions, in an air-conditioning chamber, where appropriate temperature and humidity were ensured. These are abiotic factors that can modify their composition.

*The authors suggest that "the factor limiting humans' consumption of food -based earthworms is lack of acceptance." To support this statement, it might be helpful if the authors could describe (in the background or introduction) outcomes of an animal study where earthworms replaced or substantially substituted for the normal protein source.

Thank you. Added

Current and future trends are moving towards the search for "superfoods", and earthworm biomass is nutritionally attractive. The problem is the acceptance of this unknown alternative food. Authors Conti et al. [2016] and Verneau et al. [2018] indicate that the best solution to this problem is the invisible inclusion of invertebrates (earthworms, insects) in food products.

Author Response

Response to Reviewer 3  Comments on Ref.: Ms. No. – IJERPH 2129789

Earthworm (Dendrobaena veneta Rosa) biomass as an alternative protein source

Please see our responses to the points raised by the reviewer below.

Reviewer 3

This research is very interesting and has the interest of the reviewer, although there are some points that must be addressed:

Title: The title should highlight chemical composition of earthworm biomass as its main findings

Thank you. Changed

Chemical composition of earthworm (Dendrobaena veneta Rosa) biomass is suitable as an alternative protein source

Abstract

Line 15: Define DM first before you use it as abbreviation

Thank you. Done

 Line 25: Replace “correct” with more suitable word

Thank you. Done

Introduction:

Line 38: Food security or insecurity?

Thank you. Insecurity

Line 50-51: The major factor limiting this source as food for humans (e.g., earth- 50 worms, insects) is the lack of acceptance in many European countries and its limited resources. Please include citations to support this statement.

Thank you. This had been done

Current and future trends are moving towards the search for "superfoods", and earthworm biomass is nutritionally attractive. The problem is the acceptance of this unknown alternative food. Authors Conti et al. (2018) and Verneau et al. (2016) indicate that the best solution to this problem is the invisible inclusion of invertebrates (earthworms, insects) in food products.

The argument could be more concrete by the data of “biomass production cost” of this worm or other worms that have been commercialized.

The production costs of D. veneta or E. fetida earthworm biomass depend on various factors related to the establishment and maintenance of the culture. The most important include: i) determination of the scale of the project (production on a larger or smaller scale), purchase or construction of vermireactors in which breeding will be carried out, purchase of earthworm breeding population, determination of the source of food for Lumbricidae (organic waste) - cost or no cost, determining the location of vermiculture (in countries where the air temperature does not fall below 18-20°C, you can count on intensive production of earthworm biomass all year round, otherwise, cultivation should be carried out in heated rooms for the winter period). Therefore, providing the exact costs of biomass production of these invertebrates requires an individual approach. However, it can be said with certainty that the cultivation of D. veneta or E. fetida on a household scale is not excessively cost-intensive, because it mainly boils down to the construction of vermireactors and the purchase of the initial population of earthworms. And an additional advantage of this farm (apart from the production of Lumbricidae biomass) will be the possibility of processing selected organic waste at the place of its generation into organic fertilizer (vermicompost) that can be used in plant cultivation.

Line 61-62: However, the produced earthworm biomass can be used as an alternative food source for human and animal nutrition [7]. The reference cited only highlighted on the used as animal nutrition. Please include the appropriate reference to support this statement. Plus, please include the food safety issue regarding the use of earthworm biomass as human food. Even though in times of food crisis, food safety is the most priority in food selection.

Thank you. This had been done

Conti et al. (2019) studied live E.fetida earthworms and meal made from these earthworms. Earthworms were fed with waste from the fruit and vegetable industry. The microbiological composition of fresh earthworms and earthworm meal (freeze-dried and dried) showed the absence of Salmonella spp. and Listeria monocytogenes in all samples, in accordance with Regulation (EU) 2073/2005 on microbiological criteria for foodstuffs (European Commission, 2005). The microbiological composition of fresh earthworms and earthworm meal (freeze-dried and dried) showed the absence of Salmonella spp. and Listeria monocytogenes in all samples, in accordance with Regulation (EU) 2073/2005 on microbiological criteria for foodstuffs (European Commission, 2005). In earthworms, microbiological contamination was reduced by using two different technological processes of drying for the production of meal.

Studies have shown that earthworm meal can be a safe food from the point of view of microbiological criteria. In addition, the results underlined the importance of processing methods (freeze-drying and drying) in reducing microbial contamination. As with the production of other edible terrestrial invertebrates, care must be taken to ensure appropriate processing, packaging and storage conditions to prevent microbial contamination (Conti et al. 2019).

As a further step to provide consumers with safer products, detailed guidelines should be established for the production and commercialization of earthworms if they are bred for human consumption.

Materials and Methods:

Section 2.2: My suggestion is to include a figure illustrated the vermireactors used in this study for a better view by the readers. Plus, what is the biomass production rate?

This has been added

Plus, what is the biomass production rate? The product could be expressed in kg/day.  

The scale of earthworm farming can be carried out on a different scale - small and large earthworms. Both technologies are modified by a number of abiotic (temperature, humidity) and biotic (overdensity) factors. Depending on these factors, the scale of biomass production can be determined (small-scale in g/day; large-scale in kg/day).

Line 90: mL (Please do corrections for others)

Thank you. This had been done

 Line 93: Floro-hum: pH 5.5-6.5. Composition: high moor peat, low moor peat, pearlite, sand, microelements, mineral fertilizer NPK. Please include the supplier details.

Thank you. This had been done

Line 100: remove ((pH

 Thank you. This had been done

Line 102: were presented

Thank you. This had been done

Results

Table 1: Include standard deviations for the chemical composition.

Thank you. This had been done

Discussion

Linia 179: 87,8% do

Thank you. Changed.

Line 183: indicated

Thank you. Corrected

Line 212: Please use this citation [41] to elaborate more on the food safety issue on earthworm biomass as human food.

Thank you. This had been done

Earthworm biomass has interesting nutritional properties, but in order to be commercialized as a product for human consumption and/or as animal feed, it must be safe for the final consumer. In order to ensure appropriate safety standards, the finished alternative food product should undergo a hygienic and sanitary assessment. In order to guarantee the food safety of the finished food product, it is necessary to assess the microbiological profile of the food, process hygiene indicators and additionally examine the chemical profile for the possible presence of pesticide residues and toxic elements (Tedesco et al. 2020). Therefore, an important step to provide consumers with safer products should be the establishment of specific guidelines for the production and commercialization of earthworms if they are to be bred for human consumption.

Line 248: Describe how the meals from earthworms are prepared. Explain why the people in these places could consume earthworms in their diet? What are the factors influence the acceptance and consumption in these regions

Below are recipes for dishes with the participation of earthworms. They come from 1975. They were selected from a competition for unconventional dishes, they were evaluated by a jury composed of the department of the Department of Nutrition of the Agricultural School at California State Polytechnic University in Pomona. The judges assessed each recipe for economy of ingredients, ease of preparation, and potential appeal to the eye and taste. An important factor that affects the diet is certainly the eating habits that have been shaped over the years [Gaddie i Douglas 1977].

Earthworm-based meals

EARTHWORM OMELETTE

VER DE TERRE STUFFED PEPPERS

6 eggs

3/4 to l cup fresh earthworms

1/3 cup milk, 4 cup parsley

 cup sliced celery

1/3 cup sliced green pepper

 smali onion diced

l dash Worcestershire sauce

1/3 cup shredded American cheese

Vi tsp. freshly ground pepper

Vi tsp. seasoned salt   .

l drop garlic extract

1/3 cup sliced mushroom (optional)

l drop Tabasco per egg

Beat eggs, milk, parsley, salt, pepper and garlic with a FORK until well mixed. Place mixture in medium-hot omelette pan. When half-done to taste, add worms, celery, green pepper, onion, cheese and mushrooms. Complete cooking and serve immediately.

1.) Wash 2% cups earthworms and boil 15 minutes-mod. heat. Rinse, repeat boiling, rinse and pat dry.

2.) Fry together with Vi Å‚b. of lean hamburger,*! large onion, finely chopped, 2 cloves garlic, l tsp. parsley, 1/8 tsp. pepper, V* tsp. salt and 2 smali cans tomato sauce, 6 large mushrooms thinly sliced.

3.) While you fry hamburger earthworm mixture, boil 4 to 6 beli peppers for 15 minutes or until tender.

4.) Mix into earthworm mixture l pkg. of long grain, wild rice of cracked wheat. Stuff peppers and mixture and bake at 350° for 25 minutes. Top with cheddar cheese and bakÄ™ for 5 morÄ™ minutes.

5.) Serve with chilled cream of avocado soup and carrot raisin salad.

APPLESAUCE SURPRISE CAKE

CURRIED VER DE TERRE AND PEA SOUFFLE

Vi cup  butter

1 Vi cup sugar 3 eggs

2 cups sifted flour l tsp. baking soda l tsp. cinnamon

lVi Ibs. ground earthworms

Vi cup melted butter

l tsp. grated lemon rind   .

l Vi t. salt

Vi tsp. white pepper

l egg, beaten                

l cup dry breadcrumbs,,/ l Tbs. butter

1  cup sour cream

2 Tbs. plain soda water (for lightness)

Combine earthworms, melted butter, lemon rind, salt and pepper. Stir in soda water. Shape Ä…uickly into patties. Dip patties into beaten egg and breadcrumbs. Heat butter and cook patties in it about 10 minutes, turning once. Transfer patties to hot sendng dish. Stir sour cream into skillet and heat thoroughly, pour over patties. Serve with plain boiled

potatoes.

In a saucepan heat one cup of milk and stir in 1A cup of grated fresh coconut and Vi teaspoon brown sugar. Let the mixture cool.

In V* cup butter sauce saute lightly one smali onion, grated. Add one clove of chopped garlic and cook. Stir in \Vi teaspoons curry powder and gradually add the coconut mkture. Cook the entire mixture for about 10 minutes morÄ™, then add 3 tablespoons flour mix to a paste with milk. Cook five more minutes, and allow to cool slightly.

In another pan beat 4 egg yolks well and add curry sauce gradually while stirring. Mix in one cup of drained smali cooked peas and one cup of prepared ver de terre cut in Vi inch lengths. Fold in 4 stiffly beaten egg whites, and season with salt and pepper.

Turn mixture into a buttered soufflć dish and bake souffle in a 375 oven for 40 minutes.

Conclusions

Line 264: Rephrase “general protein” with a better term

Thank you. This had been done

Line 271: “biomass of earthworms fed on segregated kitchen waste could be used in human nutrition”. This conclusion is mainly based on one previous citation [41]. It is a strong statement, therefore it should be supported based on the current finding. Please rephrase.

Further sentences have been added to support this.

Line 275: “shaping the change in eating habits among Europeans”. This study should include the consumer acceptance on earthworms for human consumption, or include this study for future work.

Thank you. This had been done

In studies evaluating the acceptance of rice wafers pure and enriched with earthworm protein, it turned out that social acceptance is mainly based on the way the information is presented on the label. Distinguishing high health and nutritional values was a factor that encouraged consumers to buy this product, which was particularly visible among men, while women behaved more conservatively. We are at the beginning of the road to introducing new food and as you can see, there is a certain consumer acceptance of such products. More time and appropriate actions are needed to convince consumers of the naturalness of products enriched with invertebrate or insect protein (Russo et al. 2020).

Round 2

Reviewer 2 Report

Thank you for your response to comments.  The manuscript reads clearer, and conclusion statements are better supported.